# Hepatic resistance to cold ferroptosis in a mammalian hibernator Syrian hamster depends on effective storage of diet-derived α-tocopherol

Daisuke Anegawa[1,2], Yuki Sugiura[3], Yuta Matsuoka[4], Masamitsu Sone[1], Mototada Shichiri[5], Reo Otsuka[1], Noriko Ishida[5], Ken-ichi Yamada [4], Makoto Suematsu[3], Masayuki Miura [2] & Yoshifumi Yamaguchi [1,6,7✉]

Mammalian hibernators endure severe and prolonged hypothermia that is lethal to non-hibernators, including humans and mice. The mechanisms responsible for the cold resistance remain poorly understood. Here, we found that hepatocytes from a mammalian hibernator, the Syrian hamster, exhibited remarkable resistance to prolonged cold culture, whereas murine hepatocytes underwent cold-induced cell death that fulfills the hallmarks of ferroptosis such as necrotic morphology, lipid peroxidation and prevention by an iron chelator. Unexpectedly, hepatocytes from Syrian hamsters exerted resistance to cold- and drug-induced ferroptosis in a diet-dependent manner, with the aid of their superior ability to retain dietary α-tocopherol (αT), a vitamin E analog, in the liver and blood compared with those of mice. The liver phospholipid composition is less susceptible to peroxidation in Syrian hamsters than in mice. Altogether, the cold resistance of the hibernator's liver is established by the ability to utilize αT effectively to prevent lipid peroxidation and ferroptosis.

[1] Hibernation Metabolism, Physiology and Development Group, Institute of Low Temperature Science, Hokkaido University, Sapporo, Hokkaido, Japan. [2] Department of Genetics, Graduate School of Pharmaceutical Sciences, The University of Tokyo, Bunkyo-ku, Tokyo, Japan. [3] Department of Biochemistry, Keio University School of Medicine, Shinjuku-ku, Tokyo, Japan. [4] Physical Chemistry for Life Science Laboratory, Faculty of Pharmaceutical Sciences, Kyushu University, Higashi-ku, Fukuoka, Japan. [5] Biomedical Research Institute, National Institute of Advanced Industrial Science and Technology (AIST), Ikeda, Osaka, Japan. [6] Global Station for Biosurfaces and Drug Discovery, Global Institution for Collaborative Research and Education (GI-CoRE), Hokkaido University, Sapporo, Japan. [7] Inamori Research Institute for Science Fellowship (InaRIS), Kyoto, Japan. ✉email: bunbun@lowtem.hokudai.ac.jp

Mammalian hibernators survive harsh seasons by invoking a depressed metabolism and a prolonged low body temperature (Tb)[1–3]. Small mammalian hibernators, including ground squirrels, chipmunks, marmots, bats, and hamsters, reduce their core Tb to below 10 °C during hibernation (HIB) (Fig. 1a). This severe hypothermic state is called deep torpor (DT) and continues for several days, sometimes over 1 week. DT is interrupted by periodic arousal (PA), in which animals leave the hypothermic state and become normothermic by rewarming through both non-shivering and shivering thermogenesis. Animals remain in the normothermic state for a period of time (typically <1 day) and then become hypothermic and immobile again by significant reduction in metabolic rate and thermogenesis. Thus, this transition between normothermia and hypothermia occurs frequently in mammalian hibernators during the HIB period (Fig. 1a), requiring metabolic reactions to constantly adjust between two temperature extremes: around 37 °C during normothermia and about 4 °C during hypothermia[4]. Two non-mutually exclusive prerequisites for enabling such an adjustment exist. One is homeoviscous adaptation, which retains adequate membrane fluidity and function at the two temperature extremes. Another is the resistance to stresses experienced during HIB, such as prolonged cold and rewarming stresses, and hypoxic/ischemic conditions[5,6].

To prepare for the above-mentioned stresses and drastic physiological changes during HIB, hibernators remodel their body systemically during the pre-HIB period[1,3]. For instance, white adipose tissues are extensively remodeled to store and supply lipids that are utilized as a fuel during HIB in not only fat-storing hibernators but also food-storing hibernators[7]. Lipid profiles change not only in white and brown adipose tissues but also in other tissues such as the heart, skeletal muscle, and liver between summer and winter and gradually during HIB, which might reflect differential utilization of lipid fuels and cellular membranous adaptation to cold[8]. In addition, seasonal remodeling of lipid composition in tissues occurs in a diet-independent manner, while there are hypotheses that dietary fatty acids, in particular, the ratio of dietary omega-3/omega-6 fatty acids, may affect the expression and quality of torpor[8–10]. In contrast, resistance to ischemia/reperfusion stresses also develops during the transition from summer to winter in 13-lined ground squirrels[11–13]. Arctic ground squirrels also undergo body remodeling, such as pre-HIB fattening and global gene expression changes[14,15], but also exhibit intrinsic resistance to global ischemia independently of seasons[16,17]. Likewise, intrinsic resistance to cold is observable not only in whole organisms, but also in cultured cells and tissues isolated from several hibernators, including ground squirrels and hamsters, regardless of seasonal and systemic body remodeling[18–23]. Several mechanisms underlying cold resistance have been proposed, including prevention of ROS generation in neurons differentiated from induced pluripotent stem cells of 13-lined ground squirrels[23], H$_2$S generation and sustained mitochondrial membrane potential in Syrian hamsters[19,22,24], and Atp5G1 gene variants specific to arctic ground squirrels[25]. However, the in vivo significance of these mechanisms is still lacking, and the molecules responsible for cold resistance remain to be elucidated.

First, why and how cold kills mammalian cells are not well understood. Cold stress induces reactive oxygen species (ROS) production associated with aberrant mitochondrial membrane potential in non-hibernator cells[22,23]. Studies using several human cancer cell lines suggested that cold-induced cell death (CICD) is similar to ferroptosis[26,27]. Ferroptosis, which was originally defined in cancer cells, is a form of non-apoptotic, iron-ion-dependent regulated cell death accompanying lipid peroxidation[28]. Its molecular mechanisms and in vivo significance continue to be intensively investigated mostly from its potential as therapeutic targets

of cancer medicine[29]. Ferroptosis sensitivity is also affected by cellular lipid profiles, at least in several human cancer cell lines. Manipulation of some metabolic enzymes of glycerophospholipids (hereafter, simply phospholipids (PLs)) or treatment with certain fatty acids changes cellular lipid composition and affects the susceptibility to ferroptosis in human cancer cell lines[30–33]. Lipid composition also affects membrane fluidity and function at different temperatures. Hence, it is of interest to compare the composition of lipids, mainly of PLs constituting cellular membranes, between non-hibernators and hibernators.

Syrian hamsters are ideal model animals for studying the mechanisms of HIB in a laboratory. The animals are classified as facultative hibernators in that they can hibernate irrespective of season if exposed to winter-like, chronic cold, and short photoperiodic conditions in a laboratory[34–36]. Syrian hamsters are food-storing hibernators that store food in their nest during the pre-HIB period and eat them in euthermic PA during the HIB period, in contrast to fat-storing hibernators that do not eat food and rely on stored body fat during HIB[7]. Under prolonged winter-like conditions, this species undergoes body remodeling from a summer-like to a winter-like body for HIB[34,37,38]. Intrinsic cold resistance has been implicated in cancer cell lines[19,22,24]. However, since these studies are based on a comparison of cancer cell lines from different origins and species, further investigation with more precise comparison is needed. In this study, using primary hepatocytes from a non-hibernator mouse and a food-storing hibernator Syrian hamster (hereafter hamster), we compared the cold sensitivity and lipid composition between them to address the mechanisms of intrinsic cold resistance in the hibernator.

## Results

**Intrinsic cold resistance of Syrian hamster hepatocytes.** We first compared the frequency of cell death in primary hepatocytes isolated from hamsters and non-hibernator mice. When incubated at 4 °C, almost all mouse hepatocytes underwent cell death within 2 days, judging by the propidium iodide (PI) uptake and large amount of lactate dehydrogenase (LDH) release (Fig. 1b, c and Supplementary Data 1). In contrast, few dying cells were observed among the hepatocytes prepared from hamsters in the non-hibernation (non-HIB) period, which were raised in summer-like conditions (Fig. 1b, c). Of note, the hamster hepatocytes exhibited the ability to survive for more than 5 days under cold culture (Fig. 1c). This remarkable cold resistance seems plausible when considering the fact that the hamsters endure DT for loner than 5 days.

Hibernating animals undergo repeated cycles of DT and PA, in which they experience transition periods of cooling and rewarming from the cold[5,6]. In vivo conditions were mimicked to address whether hepatocytes from the hamsters also possessed resistance to injuries from rewarming after the cold culture. Hepatocytes from hamsters in the non-HIB period were rapidly cooled and kept at 4 °C for 5 days and subsequently subjected to rapid rewarming at 37 °C for 24 h (Fig. 1d). As a result of this rapid protocol, 30.1 ± 11.0% of hamster hepatocytes underwent cell death after rewarming, whereas the others survived (Fig. 1e). To examine whether the speed of cooling and rewarming may affect the survival rate of hepatocytes, the effect of the slow protocol on survival was also tested (Fig. 1f). No significant differences in the survival rate were found between the rapid and slow protocols (Fig. 1g), indicating that hamster hepatocytes in vitro have resistance to cold-rewarming stress. In order to test whether the resistance to cold-rewarming stress is enhanced in HIB period, the frequency of cell death in cultured hepatocytes subjected to rapid cold-rewarming stress was compared between

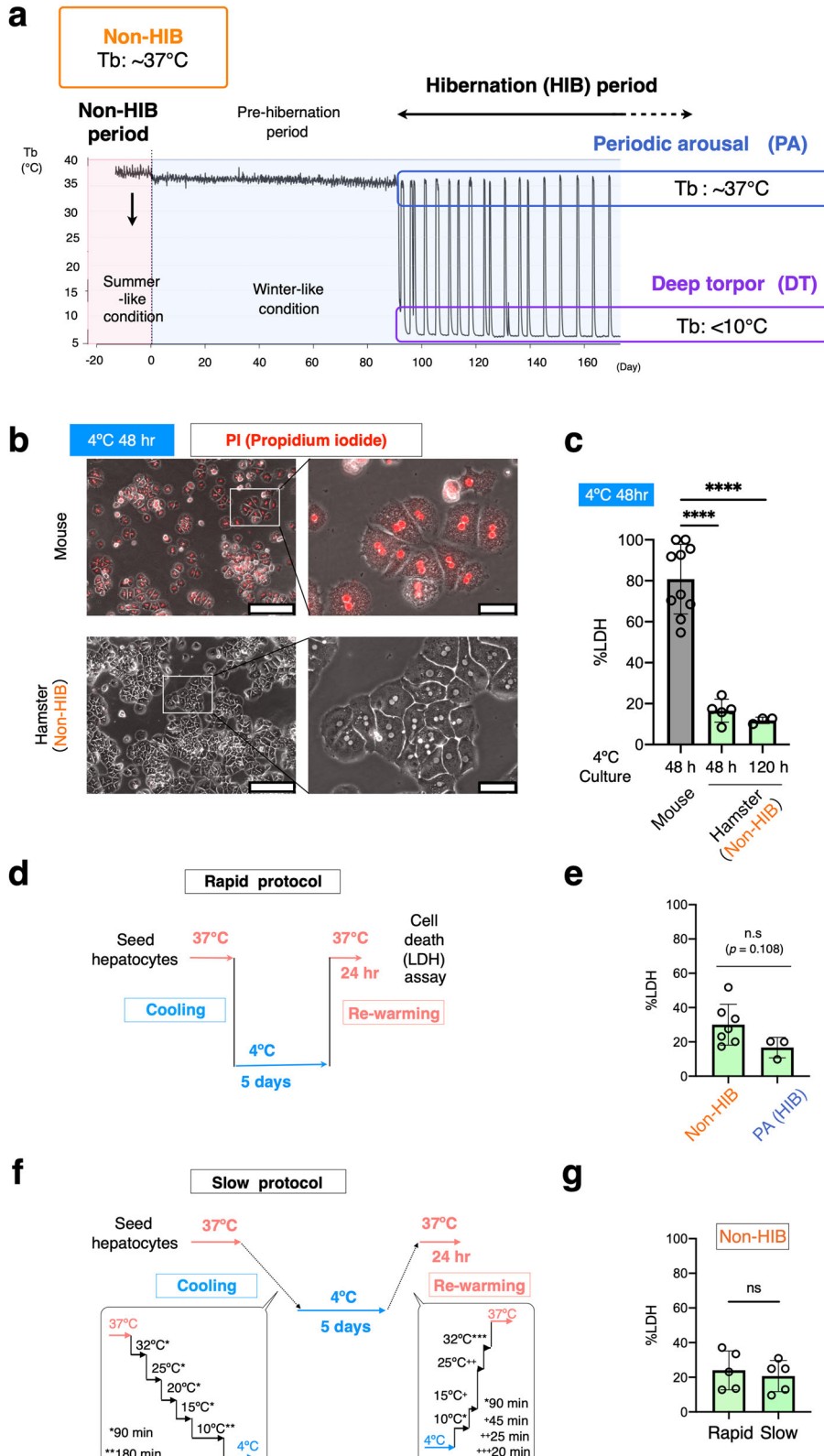

cells derived from summer-like hamsters and those from winter-like euthermic hamsters in the PA phase during the HIB period. Hepatocytes exhibited resistance to cold-rewarming stresses irrespective of HIB, although the resistance tended to be enhanced during the HIB period (Fig. 1e). The cold resistance of the hamsters was not affected by body mass of the animals used for hepatocytes isolation (Supplementary Fig. 1). These lines

of evidence indicate that Syrian hamster hepatocytes possess intrinsic cell-autonomous resistance to cold culture, regardless of HIB.

**Diet- and species-dependent cold resistance.** Over the course of exploring the mechanisms of cold resistance, we unexpectedly

**Fig. 1 Intrinsic cold resistance in primary cultured hepatocytes of Syrian hamsters. a** Schematic representation of changes in the core body temperature during hibernation in Syrian hamsters. Animals raised in summer-like (warm and long photoperiod) conditions are regarded as animals in non-HIB period. Several months after exposure to winter-like (cold and short photoperiodic) conditions, the animals begin to hibernate; the animals entered into deep torpor (DT) that lasted about 4–5 days. DT was spontaneously interrupted by periodic arousal (PA), which lasted about 12–24 h. Cycles of DT and PA were repeated continuously during HIB period. **b** Phase-contrast images of cultured hepatocytes from mouse and Syrian hamsters. Dead cells were stained with propidium iodide (PI). Scale bars; left, 250 μm. Right, 50 μm. **c** The amount of cell death determined by LDH release assay after 48 or 120 h of cold culture. ****$p < 0.0001$ (Two-tailed Welch's $t$-test). **d–g** An experimental procedure to recapitulate the DT-PA (cold-rewarming) process in cultured cells (**d**, **f**). No significant difference (ns) in the amount of cell death was found after both rapid and slow cold-rewarming stress, as determined by LDH assay, between summer-like euthermic animals and winter-like periodically aroused hamsters (**e**, **g**). Two-tailed unpaired $t$-test. Data are represented as the mean ± standard deviation (SD) and each data point in (**c**) and (**d**) represents an independent sample replicate.

---

found that the cold resistance of hamster hepatocytes diminished when certain changes were made to the hamsters' diets. Hepatocytes from hamsters raised on a standard diet (STD hamsters hereafter) exhibited remarkable cold resistance (as shown in Figs. 1 and 2a) when cultured at 4 °C, whereas those from animals raised on a stock diet (STC hamsters) did not (Fig. 2a and Supplementary Data 2). The STD and STC diets are usually used to maintain Syrian hamsters and mice in our laboratory, respectively, and differ in their ingredients (see "Methods"). It should be noted, however, that the both mice and STD hamsters used in this study were fed an STD diet. Nevertheless mouse hepatocytes did not exhibit cold resistance (Figs. 1b, c and 2d). Thus, cold resistance in hepatocytes is exerted in an STD diet-dependent manner in hamsters, but not in mice.

**Hamster's resistance to cold-induced lipid peroxidation and ferroptosis.** To gain insight into the mechanisms of diet-dependent cold resistance in hamster hepatocytes, we tried to characterize the hallmarks of CICD in hepatocytes. MitoB, a mass spectrometry probe for oxidative stress in mitochondria[39,40], was utilized to examine whether mitochondrial ROS production occurs under cold stress in hepatocytes from STC hamsters and STD hamsters. MitoB accumulates in the mitochondrial matrix, due to its cationic and lipophilic moiety and reacts with $H_2O_2$ to form a stable product, mitoP. To discriminate the cold-induced oxidation of mitoB from autoxidation during sample preparation and storage, the cold culture was conducted in the presence of $^{18}O_2$ gas in order to specifically label the mitoP generated during the cold culture (Fig. 2b). The mitoP ($^{18}O$)/mitoB ratio increased markedly under cold stress in STC hamster hepatocytes, whereas this increase was much less substantial in STD hamster hepatocytes (Fig. 2c and Supplementary Data 2). This result suggests that in STD hamsters, cold-induced mitochondrial ROS production is suppressed, and/or the produced ROS are rapidly eliminated through antioxidant mechanisms.

In non-hibernator mammals, prolonged cold stress triggers CICD, which accompanies lipid peroxidation mediated by ROS. CICD is inhibited by lipophilic radical scavengers like α-tocopherol (αT) or the iron-chelator deferoxamine in rat hepatocytes and human cancer cell lines[26,41]. These chemicals have been recently recognized as inhibitors of ferroptosis, a form of regulated cell death associated with lipid peroxidation[29]. We then tested whether CICD in hepatocytes could be prevented by ferroptosis inhibitors, including deferoxamine (an iron-chelator and antioxidant) and ferrostatin-1 and Trolox (free-radical scavengers). As predicted, these compounds effectively inhibited CICD in hepatocytes from STD mice and STC hamsters (Fig. 2d). Next, lipid peroxidation, a hallmark of ferroptosis, was examined by measuring the amount of TBARS (2-thiobarbituric acid reactive substance) in cold-cultured hepatocytes (Fig. 2e). Immediately after isolation, this measurement was not significantly different between hepatocytes from STC and STD hamsters. The

cold culture after the pre-culture significantly increased TBARS in STC hamsters, but not in STD hamsters, within 8 h (Fig. 2e), suggesting that cold-induced extensive lipid peroxidation only occurred in the hepatocytes of STC hamsters. Recent studies have proposed that specific oxidized lipids, including oxidized phosphatidylethanolamine (PE) (18:0_20:4), act as ferroptotic death sensitizers and serve as ferroptosis signatures[30,31]. Hence, we examined whether oxidized PE (18:0_20:4) was also produced during CICD. For this purpose, the $^{18}O_2$ labeling method was utilized again, using LC–MS to selectively detect the lipid peroxidation that occurred during the cold culture (Fig. 2b). This analysis demonstrated that oxidized PE (38:4) species, those corresponds with oxidized PE (18:0_20:4), were substantially produced during the cold culture by both mouse and STC hamster hepatocytes, whereas few signals indicating oxidized PE (38:4) were detected in the cold-subjected hepatocytes from STD hamsters (Fig. 2f, g). Thus, cold culture induces lipid peroxidation with a ferroptosis signature in hepatocytes from mice and STC hamsters, but hepatocytes from STD hamsters resist such ferroptosis-like lipid peroxidation.

These observations prompted an investigation into whether hamster hepatocytes were also resistant to ferroptosis-inducing agents under euthermic 37 °C conditions. In certain types of cancer cells, ferroptosis is triggered by the inhibition of glutathione peroxidase 4 (Gpx4), which reduces lipid hydroperoxide at the expense of glutathione[42]. When the ferroptosis inducers RSL3 and buthionine sulfoximine (BSO), which inactivate Gpx4 and deplete glutathione, respectively, were applied to hepatocytes, both chemicals induced a large amount of cell death in hepatocytes from STC hamsters and mice. However, hepatocytes from STD hamsters were not affected in this way (Fig. 2h and Supplementary Data 2). These results indicate that STD hamster hepatocytes resist ferroptosis as well as cold-induced ferroptosis in a diet-dependent manner.

**Hamster hepatocytes contain less highly unsaturated fatty acid (HUFA) in phosphatidylcholine (PC) and PE.** To explore the mechanisms of resistance to cold-induced ferroptosis and lipid peroxidation in hamsters, we examined the difference in lipid composition among mouse, STC hamster, and STD hamster cells, as changes in lipid profiles can affect ferroptosis sensitivity[30–32]. In fact, the amount of PE (18:0_20:4), from which ferroptosis-signature-oxidized PE (18:0_20:4) species are generated during ferroptosis, was greater in freshly isolated hepatocytes of mice than in those of hamsters (Fig. 2i and Supplementary Data 2). This raises the possibility that a higher intrinsic amount of PE-38:4 (18:0_20:4) in mouse hepatocytes might have contributed to the increase in oxidized PE (18:0_20:4) in mouse hepatocytes under cold culture (Fig. 2f, g). On the other hand, the amount of PE-38:4 (18:0_20:4) was not significantly different between hepatocytes of STC hamsters and those of STD hamsters (Fig. 2i), which did not explain the different levels of vulnerability to cold stress between the two groups. A comprehensive analysis of total

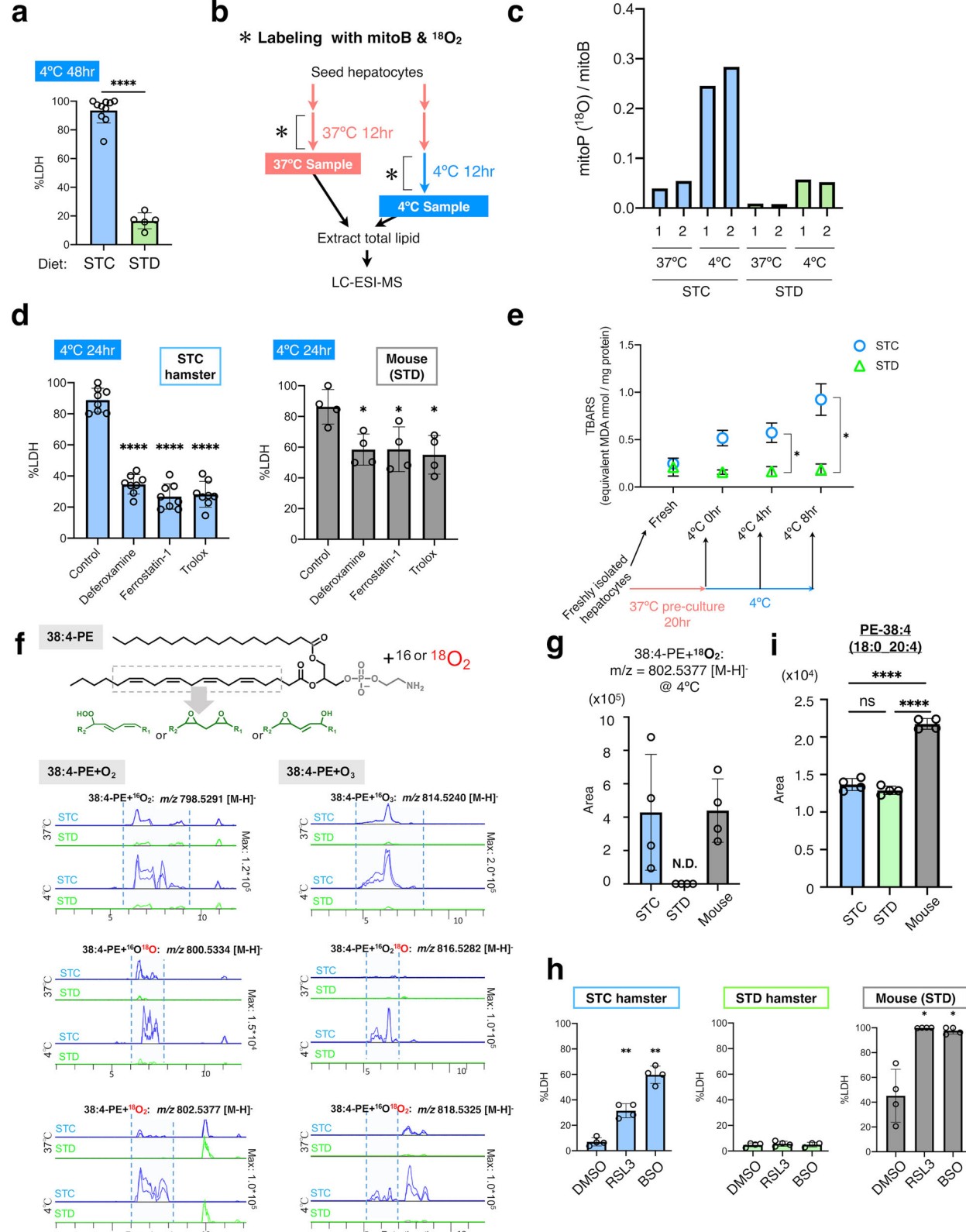

lipids extracted from freshly isolated hepatocytes was also conducted to compare two major PL classes, PL-PC, and PL-PE. Principle component analysis revealed a clear difference in the PC and PE lipid compositions between the mice and the two groups of hamsters along the PC1 axis (Fig. 3a and Supplementary Data 3), even when the mice and STD hamsters were given the same STD diet. Most lipids that largely contributed to the PC1

axis were highly unsaturated fatty acids (HUFAs) containing more than two carbon–carbon double bonds (Supplementary Table. 1). Indeed, the ratio of HUFAs to the analyzed total lipid species in the hepatocytes was greater in mice than in hamsters for both PC (Fig. 3b, c) and PE (Fig. 3b and Supplementary Data 3), indicating that hamster hepatocytes contained fewer HUFAs in their PC and PE than do mouse hepatocytes. HUFAs

**Fig. 2 Diet-dependent resistance to cold-induced ferroptosis-like cell death in Syrian hamster hepatocytes. a** Cold resistance was observed only in hepatocytes isolated from Syrian hamsters fed with the STD diet (STD), but not those fed with the STC diet (STC). ****$p < 0.0001$ (Two-tailed Welch's t-test). **b** An experimental procedure for mitoB and $^{18}O_2$ labeling to detect mitochondrial $H_2O_2$ production and lipid peroxidation under cold-culture conditions. **c** Increase in the ratio of $^{18}O$-containing mitoP to mitoB under cold culture. Results from two independent samples (1 and 2) have been shown. **d** Inhibition of cold-induced cell death by ferroptosis inhibitors (100 μM of deferoxamine, 1 μM of ferrostatin-1, or 100 μM of Trolox) in hepatocytes from STC hamsters and mice. ****$p < 0.0001$, *$p < 0.05$ versus Control (One-way ANOVA with the Tukey's multiple comparison test). **e** Lipid peroxidation (TBARS assay) occurred only in cold-cultured hepatocytes from STC hamsters. TBARS levels were measured immediately after hepatocyte isolation, after pre-culture at 37 °C for 20 h, and after cold culture for 4 or 8 h. $n = 3$ (STC) or $n = 4$ (STD) independent sample replicates. *$p < 0.05$ (One-way ANOVA with the Tukey's multiple comparison test). **f** LC–MS identification of oxidized 38:4-PE in cold-cultured hepatocytes. Upper, structure of 38:4-PE and its possible oxidized forms. Lower, comparison of extracted ion chromatogram of oxidized lipid species between STC and STD at different temperature (37 or 4 °C). Multiple peaks for oxidized 38:4-PE species (inside dashed lines) dominantly appeared in STC hepatocytes and those labeled with $^{18}O$ were detected only in STC hepatocytes at 4 °C. **g** Relative quantification of PE (38:4) + $^{18}O_2$, an oxidized PE (38:4) with two $^{18}O$ atoms. It was detected by LC–MS analysis in hepatocytes from STC hamsters or mice, but not from STD hamsters, after 12-h cold culture. N.D. (not detected). **h** Ferroptosis was induced by 2-μM RSL3 or 1-mM BSO in hepatocytes from STC hamsters and mice, but not from STD hamsters, under 37 °C culture conditions. **$p < 0.01$, *$p < 0.05$ versus Control (One-way ANOVA with the Tukey's multiple comparison test). **i** The amount of PE (38:4) in freshly isolated hepatocytes measured by LC–MS analysis. ****$p < 0.0001$, ns $p > 0.05$ (One-way ANOVA with the Tukey's multiple comparison test). Data are represented as the mean ± SD and each data point in (**a**), (**b**), (**g**), (**h**) and (**i**) represents an independent sample replicate.

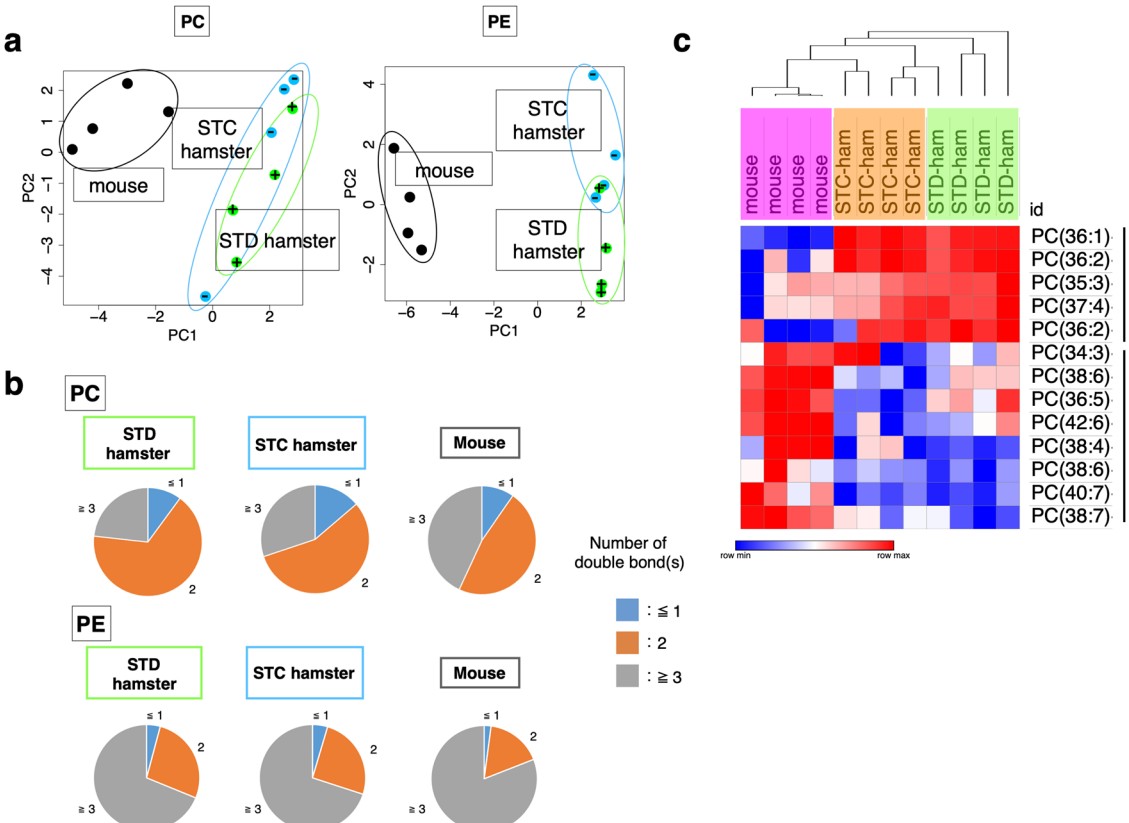

**Fig. 3 Lipidome analysis revealed different phospholipid compositions of hepatocytes between mouse and hamster. a** Principle component analysis (PCA) of PL-PC and PL-PE species in freshly isolated hepatocytes from mouse, STC hamster, and STD hamster. **b** Pie charts showing the average percentage of lipid species classified by the number of carbon–carbon double bonds. **c** Hierarchical clustering of the relative amount of PC species among distinct groups of animals. Relative abundance of each lipid species among samples is shown by blue red gradient.

are targets of oxidation and become the origin of lipid peroxidation. Taken together, differences in the amounts of HUFAs, including PE (38:4), between hamsters and mice may contribute to the difference in cold-induced oxidative stress sensitivity between hepatocytes of these two species.

**Diet-derived αT confers cold resistance on hamster hepatocytes.** Finally, we determined which nutrient in the STD diet was responsible for granting cold resistance to hamster hepatocytes. Although the amounts of many ingredients differ between the

STD and STC diets, the difference in the amount of the dietary antioxidant vitamin E is notable. According to the manufacturer's information, the STD diet contains five times more vitamin E than that found in the STC diet (STD: 208.3 mg/kg, STC: 45.2 mg/kg). Vitamin E is a lipophilic radical scavenger, acting as a chain-breaking antioxidant to prevent lipid peroxidation and ferroptosis. Therefore, a rescue experiment was conducted to test whether a high intake of dietary vitamin E was sufficient for granting cold resistance to cold-vulnerable hepatocytes from STC hamsters. An appropriate amount of αT, a major form of the vitamin E analogs, was estimated as 20-μg/g body mass per a day

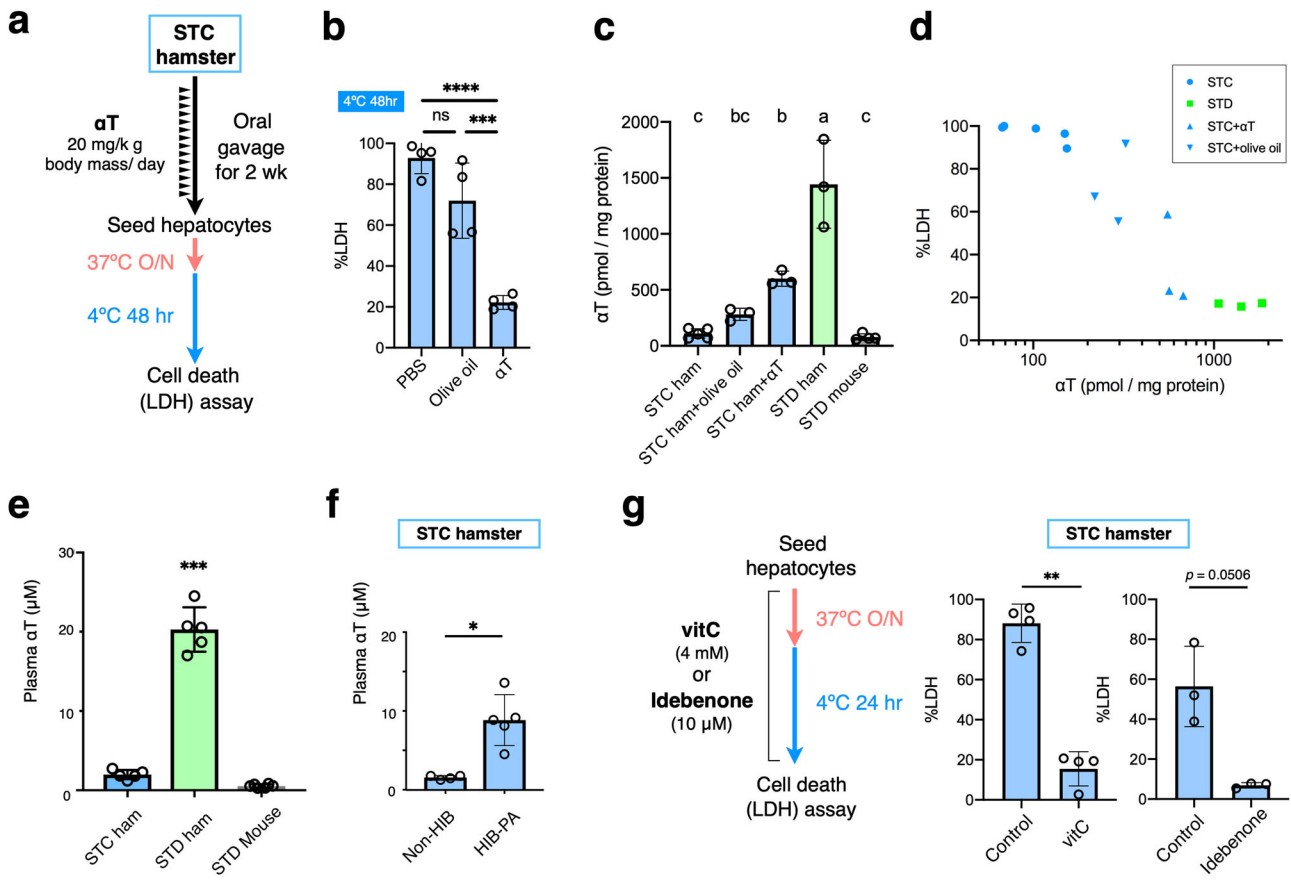

**Fig. 4 Intake of sufficient dietary αT is required for the cold resistance of hamster hepatocytes. a** Schemes for oral administration of αT to hamsters. 20-μg αT/g body mass dissolved in olive oil was administered to STC hamsters once a day for 2 weeks. **b** The amount of cell death after 48-h cold culture among hamsters administered with PBS, olive oil, or αT. ****$p < 0.0001$, ***$p < 0.001$, ns $p > 0.05$ (One-way ANOVA with Tukey's multiple comparison test). **c** αT content in freshly isolated hepatocytes. αT level was normalized to the total protein amount in the hepatocyte lysates. Letters a, b, c refer to the significant differences of αT level from each other—if two columns do not share a letter, they are significantly different ($p < 0.05$) (One-way ANOVA with the Tukey's multiple comparison test). **d** Relationship between the amount of cell death and αT content in (**c**). **e, f** αT concentration in plasma of STC hamsters, STD hamsters, and STD mice in summer-like (non-HIB) condition, and of STC hamsters in summer-like, non-HIB, or at periodic arousal during HIB (HIB-PA). ***$p < 0.001$ (One-way ANOVA with the Tukey's multiple comparison test), *$p < 0.05$ (Welch's $t$-test). **g** Inhibition of cold-induced cell death by 4-mM vitamin C or 10-μM idebenone in STC hamster hepatocytes. **$p < 0.01$ (Two-tailed paired $t$-test). A scheme for treating hepatocytes with vitamin C or idebenone is shown. Data are represented as the mean ± SD and each data point in (**b–g**) represents an independent sample.

and orally administered to STC hamsters for 2 weeks, and their hepatocytes were subsequently subjected to cold culture. As a result, CICD was significantly suppressed in hepatocytes from αT-treated hamsters compared with those from the vehicle control, olive-oil-treated hamsters (Fig. 4a, b and Supplementary Data 4). Thus, the cold resistance of hamster hepatocytes is reliant on a sufficient amount of dietary αT. To examine how dietary αT contributed to cold resistance in hamster hepatocytes, αT content was measured in freshly isolated hepatocytes[43]. This analysis revealed that hepatocytes from αT-treated hamsters contained a higher amount of αT compared with those from olive-oil-treated hamsters or non-treated STC hamsters (Fig. 4c). STD hamsters exhibited a much higher αT content and lower frequency of cell death than that exhibited by αT-treated STC hamsters, and the preventive effect on cell death correlated with the αT content of hepatocytes (Fig. 4c, d and Supplementary Data 3). Interestingly, the amount of αT in mouse hepatocytes was about 1/10th of that in hamster hepatocytes, despite the fact that the mice were fed with the same STD diet as the STD hamsters (Fig. 4c). Plasma αT concentration was also higher in the STD hamsters than in the mice fed with STD diet or the STC hamsters (Fig. 4e), which is consistent with the fact that most of

αT absorbed from diets is redistributed via liver into whole body[44]. Thus, hamsters, but not mice, have the capacity to retain a high amount of diet-derived αT in their hepatocytes as well as in the blood circulation, which contributes to the cold resistance of hamster hepatocytes by preventing lipid peroxidation. Interestingly, the plasma αT concentration of euthermic STC hamsters at PA during the HIB period was about fivefold higher (8.85 ± 3.23 μM) than those in the non-HIB period (1.57 ± 0.24 μM) (Fig. 4f), implying the existence of systems to compensate for the low amount of αT taken from the diet during the HIB period in Syrian hamsters.

αT is regenerated from its oxidized form, αT radical, by vitamin C or ubiquinol (Coenzyme $QH_2$) in a distinct chemical reaction[45–47]. Given that αT is consumed to prevent lipid peroxidation during cold culture, it is expected that cold-vulnerable STC hepatocytes containing small amounts of αT may exhibit improved cold resistance when αT regeneration is enhanced. Consistent with this idea, the addition of vitamin C or idebenone, a Coenzyme $QH_2$ analog, effectively inhibited CICD in hepatocytes from STC hamsters (Fig. 4g). Taken together, these data suggest that the cold resistance of hamster hepatocytes is established via retention of a large amount of αT, which is utilized to prevent lipid peroxidation.

## Discussion

One of the greatest and longest-lasting mysteries of HIB is how mammalian deep hibernators endure severe hypothermia (<10 °C) for several days to over a week and the stress that results when the Tb transitions from 4 to 37 °C within a few hour during PA[3,4]. Such a large and rapid transition of Tb is a consequence of drastic changes in metabolic rate, which can cause ROS generation and oxidative stress[5]. To maintain membrane fluidity and function at very low temperatures, a highly unsaturated PL profile, in which polyunsaturated fatty acids (PUFAs)-PLs are enriched, is preferable[48]. However, at the same time, such a highly unsaturated PL profile is vulnerable to oxidative stress during PA, as the carbon–carbon double bonds in PUFAs, and especially in HUFAs, are more easily oxidized than those in MUFAs due to either a chain reaction of lipid peroxidation or oxygenases such as LOX and COX[49]. Thus, the capacity to hibernate demands the prevention of lipid peroxidation and oxidative stress during DT and PA[4,50]. This study demonstrated that a mammalian food-storing hibernator, the Syrian hamster, effectively stored and utilized diet-derived αT in the liver and plasma to scavenge generated ROS, lipid peroxidation, and cold-induced ferroptosis, all of which can occur in severe hypothermic DT and repeated PA during HIB. Since αT and other vitamin E analogs act as chain-breaking antioxidants to prevent lipid peroxidation and are easily obtained from available foods in wild environments[50], maintaining higher level of αT in the liver and the plasma seems to be an ideal solution to these challenges of HIB, particularly for food-storing hibernators that can ingest dietary αT during PA. Further studies will be needed to examine whether fat-storing hibernators such as ground squirrels also have superior hepatic capacity for αT storage.

To the best of our knowledges, this study is also the first to clearly demonstrate that differences in PL composition of liver exist between a non-hibernator (mouse) and a hibernator (Syrian hamster). Although previous studies using GC–MS have determined fatty acid proportions in hibernators-fed different diets[51], few studies have been reported that used a similar instrument and method to compare lipid compositions between non-hibernators and hibernators that had been maintained under the same diet and laboratory conditions. Such a comparison is quite important because lipid composition can be affected by various factors such as diet, ambient temperature, seasonal body remodeling, methods of sample preparation and detection, etc.[8,52,53]. Our approach here led to the unexpected finding that the compositions of hepatic HUFAs including PC and PE were much higher in the mouse than in the Syrian hamster when fed with the same diet in the same laboratory conditions. This difference in the PLs composition may contribute to the greater susceptibility of the non-hibernator mouse to severe cold stress than that of the Syrian hamster. Specifically, the susceptibility to ferroptosis inducers in some human cancer cell lines is reduced by suppression of acyl-CoA synthase 4, which synthesizes long-chain polyunsaturated CoAs with a preference for arachidonic acid (C20:4 n-6), or of lysophosphatidylcholine acyltransferase 3, which remodels PLs via the reacylation (Lands) cycle. Suppression of these enzymes results in a decrease of PE (18:0_20:4) containing arachidonic acid, and it is known that oxidized PE (18:0_20:4) increases susceptibility to ferroptosis inducers[30–32]. As the amount of PE (38:4), a possible source of PE (18:0_20:4), was lower in the hepatocytes of Syrian hamsters than in those of mice irrespective of diet, such a unique PL composition in Syrian hamsters might play a part in rendering the cells resistant to ferroptosis. Thus, this study provides evidence that hepatic CICD in the mouse and the Syrian hamster can be regarded as ferroptotic cell death, judging by the hallmarks of ferroptosis, including necrotic morphology, lipid peroxidation, generation of oxidized PUFA

[PE-38:4(18:0_20:4)] (a known ferroptosis signature), and inhibition of cell death by deferoxamine (an iron chelator) and αT (a lipophilic radical scavenger). Pathophysiological conditions such as acute kidney injury, heart transplantation, neurodegenerative disorders, and nonalcoholic steatohepatitis are found to accompany ferroptosis[54,55]. The findings herein may lead to a new therapeutic approach to preventing tissue damage in human medicine by maintaining a high content of αT within non-hibernator cells.

One limitation of this study was that the importance of αT in HIB was not directly assessed in vivo. Although this study demonstrated that hepatocytes from hamsters fed with an STC diet were vulnerable to cold-induced ferroptosis under cultured conditions, we observed that the STC hamsters hibernated and survived (our unpublished observation). The discrepancy between the in vitro cold vulnerability and the in vivo cold resistance may be explained by the existence of mechanisms that compensate for the hepatic αT shortage during HIB. One possible compensatory mechanism is that antioxidants, including vitamin C and vitamin E itself, are supplied systemically from other organs via the blood stream. In fact, we found that the STC hamsters in the HIB period exhibited much higher plasma αT concentrations than those in summer-like conditions. In addition, it has been reported that the plasma concentration of vitamin C, which can regenerate αT radicals, changes markedly during cycles of PA and DT bouts in arctic ground squirrels and Syrian hamsters[56–58]. In rodents, vitamin C is synthesized in the liver and other organs and may be supplied to the liver during HIB. As such, a continuous supply of vitamin C from the blood may regenerate αT, thereby preventing its depletion and lipid peroxidation in cells and tissues during HIB. Thus, retention and regeneration of αT may prevent cell death in a concerted manner in vivo during HIB. This idea is also consistent with a previous report that plasma αT concentration increased during HIB in Syrian hamsters[58]. Although it was implicated that high amount of dietary αT may inhibit torpor in golden mantled ground squirrels[59], more intervention-driven studies using a large number of animals with time-course sampling during HIB, ingredient-precisely controlled diets, and genetic manipulation will be necessary to evaluate the role of effective storage of αT in HIB physiology[3]. Another limitation of this study is that it is unclear at present whether the action of αT is parallel to or involved in in vitro cell-autonomous cold resistant mechanisms that have been proposed in other cell types and cancer cell lines derived from hibernators[19,22,23,24,25]. Since the in vivo significance of these mechanisms is also lacking, future studies on the mechanistic dissection of their functional contribution to and crosstalk in HIB physiology are needed.

In conclusion, this study has identified that the intrinsic property of maintaining a high αT content in the hepatocytes underlies the mechanism of cold resistance in a mammalian food-storing hibernator, the Syrian hamster, which will be of help in understanding the long-lasting mystery of cold resistance in mammalian hibernators and its application to medical field.

## Methods

**Animals and housing.** Male Syrian hamsters (*Mesocricetus auratus*) from an outbred colony were purchased from SLC, Inc, Japan. Animals were housed in groups of 3–4 animals per cage with ad libitum access to diets (MR standard (STD) diet or MR stock (STC) diet, Nihon Nosan, Japan) (Supplementary Table 2) and water under summer-like conditions (light condition = 14L:10D cycle, lights on 06.00–20.00, ambient temperature = 22–25 °C) for over 2 months. For HIB induction, 8- or 9-week-old hamsters were purchased, and subsequently reared under winter-like conditions (8L:16D cycle, lights on 10.00–18.00, ambient temperature = 5 °C). Animals were individually housed in polypropylene cages. The onset of HIB was detected comprehensively by the characteristic postures of animals (rolled into ball), reduced activity, and consumption of food when cages were changed. The sawdust method was also used for confirming that animals successfully hibernated; wood chips placed on the back of hibernating individuals

remained in place until the animals experienced a PA[36]. Under this condition, most animals start HIB 2–4 months after cold exposure[34]. For sampling animals in the PA phase during the HIB period, we observed animal status every morning by visual inspection with the sawdust method mentioned above to check whether the hibernating animals were in the PA or DT phase. Euthermic animals in spontaneous PA phases, judged by locomotion in cages and the reaction to handling, were anesthetized and used for hepatocyte culture around 12:00–16:00 (ZT2-6) (see isolation of hepatocytes for details). In this study, Tb and duration during PA were not determined, as a Tb logger (iButton) was not implanted into the animals used for hepatocyte culture because of the concern that surgical operation of Tb loggers into the body cavity might affect liver physiology and hepatocyte culture. Animals were sacrificed under anesthesia with 4.5% isoflurane for hepatocytes isolation by reperfusion or for blood collection by decapitation at 13–18 weeks of age, except for Figs. 1d, e and 4f in which the animals were at 31–36 weeks of age.

Male C57BL/6 mice were purchased from SLC, Inc, Japan. Animals were reared under summer-like conditions (light condition = 14L:10D cycle, lights on 06.00–20.00, ambient temperature = 22–25 °C) and had ad libitum access to STD diet (MR standard diet, Nihon Nosan, Japan) and water in this experiment. Because we did not have information on the diet the mice had been fed in the breeding company, we purchased the mice at 8 weeks of age and fed the STD diet for over 2 months and used for hepatocyte culture experiments at 16–21 weeks of age.

All animal care and experimental procedures were approved by the Ethics Committees of the University of Tokyo (Ethical Approval no. P28-11) and Hokkaido University (Ethical Approval no. 18-0140), and conducted according to the ethics guidelines of the University of Tokyo and Hokkaido University.

**Chemicals**. Reagents used in this study are as follows; Deferoxamine (Sigma Aldrich, D9533), Ferrostatin-1 (Sigma Aldrich, SML0583), Trolox (Cayman Chemical, 10011659), vitamin C (Ascorbic acid) (Wako, 012-04802), RSL3 (Sigma Aldrich, SML2234), BSO (Cayman Chemical, 14484).

**Isolation of hepatocytes**. Hepatocytes from the livers of hamsters and mice were prepared by the two-step collagenase perfusion technique, according to previous studies with minor modifications[60]. Briefly, after anesthesia with 4.5% isoflurane, the livers were exposed and perfused with a solution containing 1-mM EGTA (ethylene glycol tetraacetic acid) in $Ca^{2+}/Mg^{2+}$-free Hank's balanced salt solution for about 5 min, followed by a solution containing 1-mg/mL collagenase with $Ca^{2+}/Mg^{2+}$ in Hank's balanced salt solution for 2–3 min at a flow rate of 13 mL/min for hamsters and 6 mL/min for mice. The livers were gently removed and put into EMEM (Sigma Aldrich, M4655) containing 10% FBS. Hepatocytes were obtained after mechanical dissociation by shaking the liver tissue gently in the EMEM, grabbing with tweezers with a circular tip, and subsequently filtering with a 100-μm mesh cell strainer (TOKYO SCREEN CO., LTD). The cells were collected via centrifugation at $40 \times g$ for 1 min. Then, the cells were resuspended in 24.5 mL of complete Percoll medium (composed of 12.5 mL of L-15 [Gibco 11415-064] medium supplemented with 0.429-g/L HEPES, 2-g/L BSA [Sigma A1470], $1 \times 10^{-7}$-M insulin [Wako 093-06471], 1.2 mL of 10× HBSS(-) and 10.8 mL of Percoll [GE Healthcare 17-5445-02]), and live parenchymal cells were purified via centrifugation at $60 \times g$ for 10 min. Then, the cells were washed twice in EMEM containing 10% FBS via centrifugation at $40 \times g$ for 2 min, and finally filtered with a 40-μm cell strainer (BD Falcon). Cell viability (>80%) was assessed by the trypan blue exclusion test.

**Preparation of collagen-coated plate**. Atelocollagen acidic solution (KOKEN, IPC-30) was diluted at 0.5% in PBS. The diluted collagen solution was added to tissue culture-treated plates (Corning) and incubated 5 min at room temperature. The plates were washed with PBS once and placed at 37 °C until they were used.

**Primary culture of hepatocytes**. The basal medium for culturing hepatocytes was DMEM/F12 (Gibco 21041-025) supplemented with 5-mM HEPES (Gibco 15630-080), 30-mg/L L-proline (Wako 161-04602), 0.5% BSA (Sigma A1470), 10-ng/mL epidermal growth factor (Sigma E4127), 1% Insulin, Transferrin, Selenium, Ethanolamine Solution (ITS-X) (Gibco 51500056), $1 \times 10^{-7}$-M dexamethasone (Wako 047-18863), 10-mM nicotinamide (Wako 141-01202), 1-mM L-ascorbic acid 2-phosphate (Wako 323-44822), and 1% Penicillin-Streptomycin (Wako 168-23191). Hepatocytes were seeded on collagen-coated plates and cultured at 37 °C in a 5% $CO_2$ incubator. A 10% FBS-supplemented basal medium was used as the seeding medium. Three to four hours after seeding, the medium was replaced with serum-free basal medium, and the cells were cultured for about 16 h for stabilization. Then, cold culture was conducted in a 4 °C refrigerator using the basal medium containing 100-mM HEPES (pH 7.4). This formulation is sufficient for keeping a pH of 7.5 at 4 °C, excluding the possibility that pH dysregulation affects cell viability. For the slow protocol, additional refrigerators and another cell culture incubator were set to 10, 15, 20, 25, and 32 °C, respectively, and cell culture dishes were transferred successively from the incubators to refrigerators.

**Cell death assay**. Hepatocytes were incubated with 1-μg/mL PI (Sigma P4170) for about 30 min and observed under a DMi8 microscope (Leica Microsystems). The amount of LDH in the supernatant was measured using an LDH cytotoxicity detection kit (Takara MK401) following the manufacturer's instructions. The amount of LDH in each sample was normalized against that of hepatocytes fully lysed with Triton X-100 (final 1% in PBS). The samples were individually mixed with reagents on microplates, and the absorbance was measured at 490 and 630 nm (as a control) using 2030 ARVO™ X (Perkin Elmer) or Multiskan GO (Thermo Fisher Scientific) after a 30-min incubation at room temperature.

**$^{18}O_2$ labeling**. Three hours after seeding the hepatocytes, the culture medium was changed to the new basal medium containing 5-μM mitoB (Cayman Chemical, 17116). To replace oxygen with $^{18}O_2$ as much as possible, a bubbling treatment was carried out with $^{18}O_2$-containing gas in the new basal medium [Gas A (N$_2$: 75%, $^{18}O_2$: 20%, $CO_2$: 5%) for 37 °C culture and Gas B (N$_2$: 80%, $^{18}O_2$: 20%) for 4 °C culture]. Gas mixing was completed by a custom-made gas mixer. After the medium change, the cell culture plates were put into a chamber that had been incubated at 37 °C in advance. Then, the chamber was filled with $^{18}O_2$-containing gas for 30 min at a flow rate of 20 cc/min, which was presumed to replace the gas in the chamber completely. After that, each chamber was placed in a 37 °C incubator or a 4 °C refrigerator and cultured for 12 h. Then, the cells were washed with PBS once and homogenized in 1-mL MeOH ($5 \times 10^5$ cells/mL). These homogenates were stored at −80 °C and used for measurements of mitoP/B[39,40] and lipids within 2 weeks.

**Measurement of mitochondrial hydrogen peroxide in vitro**. We prepared MitoB ($C_{25}H_{23}BBrO_2P$, molecular weight 477.14, Sigma) stock solution in sterile saline. Diluted MitoB solution (0.5 mg/mL in a 1:1 solution of DMSO:PBS) was added to the hepatocyte culture. The production of mitochondrial hydrogen peroxide during incubation was assessed by determining the MitoP/MitoB ratio using LC–MS/MS (LC-MS8040, Shimadzu Corporation), by monitoring positive ion transitions of $m/z$ 397.1 > 183.0 and $m/z$ 369.1 > 183.0 for MitoB and MitoP, respectively[40]. The samples were resolved on the LUNA Phenyl-Hexyl column (100 × 2.0 mm I.D × 100 mmL, 3-μm particle, Shimadzu GLC), using a step gradient with mobile phase A (0.1% formate) and mobile phase B (0.1% acetonitrile) at ratios of 90:10 (0–2 min), 65:35 (2–5 min), 50:50 (5–10 min), 0:100 (10–14 min), and 90:10 (14–20 min), at a flow rate of 0.2 mL/min and a column temperature of 40 °C.

**TBARS assay**. Lipid peroxidation levels in the hepatocytes were measured by a TBARS (TAC method) assay kit (Cayman Chemical). For the fresh samples, $1 \times 10^6$ hepatocytes were washed immediately after isolation with PBS ($40 \times g$ or $100 \times g$, 2 min) and homogenized with 200 μL of homogenizing buffer (50-mM phosphate buffer, 1-mM EDTA, 1% Triton X-100). For the 4 °C samples at 0, 4, and 8 h, $1 \times 10^6$ hepatocytes cultured on 60-mm dishes were washed with PBS after 37 °C pre-culture or cold culture and homogenized with 200 μL of homogenizing buffer. These homogenates were stored at −80 °C until use. A TBARS assay was conducted using 100 μL of these homogenates following the manufacturer's instructions. Briefly, the homogenates were mixed with reagents and incubated at 98 °C for 1 h. After being incubated for 10 min on ice, the samples were centrifuged at $1600 \times g$ for 10 min at 4 °C. Then, absorbance was measured at 540 nm using Multiskan GO (Thermo Fisher Scientific). The same homogenates were used to quantify the total protein amount by the BCA method, and the TBARS level was normalized to the protein amount.

**Lipidome analysis and identification of lipid peroxidation with LC–MS**. Freshly isolated hepatocytes were centrifuged and the pellets were stored in 1.5-mL Eppendorf tube at −80 °C until use. For total lipid extraction, 100 μL of 1-butanol/methanol (1:1, v/v) with 5-mM ammonium formate was added into the pellets, and the mixture was vortexed for 10 s, sonicated for 15 min in a sonic water bath, and then centrifuged ($16,000 \times g$, 10 min, 20 °C). The supernatant was transferred into a 0.2-mL glass insert with Teflon insert cap for analysis by LC ESI-MS[61].

For lipidomic analysis, an orbitrap type MS (Q-Exactive focus, Thermo Fisher Scientific, San Jose, CA), that enables us to perform highly selective and sensitive metabolite quantification owing to the Fourier Transfer MS principle, was connected to a HPLC (Ultimate3000 system, Thermo Fisher Scientific). LC and MS conditions were based on Růžička et al.[62]. Briefly, the samples were resolved on the Thermo Scientific Accucore C18 column (2.1 × 150 mm, 2.6 μm) with mobile phase A (10-mM ammonium formate in 50% acetonitrile (v) and 0.1% formic acid (v)) and mobile phase B (2-mM ammonium formate in acetonitrile/isopropyl alcohol/water, ratios of 10:88:2 (v/v/v) with 0.02% formic acid (v)), using step gradient at ratios of 65:35 (0 min), 40:60 (0–4 min), 15:85 (4–12 min), 0:100 (12–21 min), 0:100 (21–24 min), 65:35 (24–24.1 min) and 100:0 (24.1–28 min), at a flow rate of 0.4 mL/min and a column temperature of 35 °C.

The Q-Exactive focus mass spectrometer was operated under an ESI positive and negative mode. Full mass scan ($m/z$ 250–1100) followed by three rapid data-dependent MS/MS, was operated at resolutions of 70,000 and 17,500, respectively. The automatic gain control target was set at $1 \times 10^6$ ions, and maximum ion injection time was 100 ms. Source ionization parameters were as follows; spray voltage at 3 kV, transfer tube temperature at 285 °C, S-Lens level at 45, heater

temperature at 370 °C, Sheath gas at 60, and auxilliary gas at 20. Acquired data were analyzed by Qual browser for oxidized lipid analysis, and by LipidSearch software (Mitsui Knowledge Industry, Tokyo, Japan) for major PLs with following parameters; search parameters: precursor mass tolerance = 3 ppm, product mass tolerance = 7 ppm, m-score threshold = 3.

**Oral administration of αT.** The Syrian hamsters ate about 385-g/kg body mass of diet per week in our laboratory. According to the manufacturer's information, the difference in daily intake of vitamin E between the STD diet and the STC diet was estimated to be about 9.0-μg/g body mass. To compensate for this difference and avoid overdose, αT (Tokyo Chemical Industry, T2309) dissolved in olive oil (Wako, 150-00276) at a concentration of 20 μg/μL was administered at 1-μL/g body mass to hamsters fed with the STC diet for more than a month. As controls, olive oil or PBS were given to other groups of animals. Administration of αT or the controls was conducted by oral gavage once daily between 11:00 and 13:00 for 14 consecutive days before hepatocytes were cultured from the animals.

**Quantification of αT with high-performance liquid chromatography (HPLC).** Freshly isolated hepatocytes were washed with PBS and stored at −80 °C until αT extraction. Plasma was obtained by centrifuging blood immediately after the collection from a finger of hamster or a tail of mouse using heparin-coated capillary tubes. 1/100 volume of 13% EDTA-2K was added to the collected blood immediately and mixed, and then the blood was centrifuged at $1200 \times g$ for 10 min at room temperature. The supernatant was collected as plasma, immediately frozen in liquid $N_2$, and stored at −80 °C until use. The collected blood was extraction and measurement of αT was conducted as per a previously described method[43]. Briefly, cell pellets or plasma were homogenized using a bead mixer with PBS, and lysates were used for the protein quantification with a BCA assay and for vitamin E measurement. Extraction of total lipids from the lysates or plasma was done by vigorously mixing them with chloroform/methanol (2:1 by volume) containing 100 μM of butylated hydroxytoluene to avoid the in vitro oxidation. Hepatic αT contents were measured using HPLC with an amperometric electrochemical detector (ECD) (Nanospace SI-1; Osaka Soda, Japan) set at 700 mV, with an ODS column (Wakosil-II 5C18 RS, 5 μm, 250 × 4.6 mm; Fuji film Wako, Osaka, Japan, or) followed by a reducing column (RC-10, 15 × 4 mm; Shiseido) with methanol containing 50-mM sodium perchlorate as eluent at 0.7 ml/min. αT in plasma was measured using HPCD-ECD equipped with a reducing column (ECD-700, Eicom, Japan). Methanol containing 50-mM sodium perchlorate, 2% of water, and 0.1% sodium acetate was used as eluent.

**Statistics and reproducibility.** Statistical analyses were conducted using Graph-Pad Prism unless otherwise mentioned. We repeated same experiments at least two and typically over three times, except data on PL metabolomics (Fig. 3) and mitoP/B labeling (Fig. 2c) due to cost limitation and restriction. Each sample size is indicated in the figure or legend.

**Reporting summary.** Further information on research design is available in the Nature Research Reporting Summary linked to this article.

## Data availability
Source data underlying figures are presented in Supplementary Data 1–4. All other data are available upon reasonable request to the corresponding author Y.Y.

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

## Acknowledgements

We are grateful to T. Miyake, H. Kusuhara, and Y. Ito for instructing the primary hepatocyte cultures, and N. Kono, S. Shuji, and all members of Miura's and Yamaguchi's laboratories for helpful discussion. We also thank Y. Chayama, L. Ando, Y. Sato, T. Fujimoto, H. Taii, H. Makino, K. Sone, and S. Enju for experimental assistance. This work was supported by grants from the Japan Science and Technology Agency (JPMJPR12M9 to Y.Y.) and from the Japanese Society for the Promotion of Science, and the Ministry of Education, Culture, Sports, Science, and Technology in Japan (JP20H05766, JP19H04046, and JP18K19321 to Y.Y., JP18K14884 to Y.M., JP17H03977 and 18K19405 to K.Y., and JP16H06385 to M.M.). This work was also supported by grants from the Japan Agency for Medical Research and Development (AMED) to Y.Y. (JP20gm6310019), M.M. (JP17gm061004 and JP19gm5010001), and K.Y. (JP19gm0910013), by a grant from AMED Moonshot R&D Program to Y.S. (21zf0127003h001), and by grants to Y.Y. from the Cell Science Foundation, Sekisui Chemical Innovations Inspired by Nature Research Support Program, the Sumitomo Foundation for Basic Research Projects, the Takeda Science Foundation, Toray Science foundation, the Naito Foundation, the Uehara Memorial Foundation, the Mochida Memorial Foundation for Medical and Pharmaceutical Research, Terumo Life Science Foundation, Inamori Foundation, and NIBB Collaborative Research Program (16-408 and 20-101). D.A. was a research fellow of the Japan Society for the Promotion of Science.

## Author contributions

D.A., Y.S. and Y.Y. conceived the experiments. D.A., Y.S., Y.M., M. So., N.I., M. Sh., R.O. and Y.Y. performed the experiments. D.A., Y.S., Y.M., M. So., R.O. and Y.Y. analyzed data and wrote the manuscript. M.M., K.Y., M. Su. and Y.Y. supervised the study.

## Competing interests

The authors declare no competing interests.
