## [Peer Review File · Communications Biology]

Reviewers' Comments:

Reviewer #1:

Remarks to the Author:

In this study, the authors investigated the mechanisms responsible for cold resistance in Syrian hamster's hepatocytes, which exhibited remarkable resistance to prolonged cold culture. In contrast, hepatocytes of non-hibernating mice underwent cold-induced cell death showing hallmarks of ferroptosis, such as lipid peroxidation. This study further revealed a diet-dependent resistance of Syrian hamster's hepatocytes to cold- and drug-induced ferroptosis linked to the specific retention of alpha-tocopherol in the liver. Below the authors can find some major and other minor comments to help them improve their manuscript.

Major comment:

- One major comment resides in the status of the introduction, which is actually missing important and key information necessary for the understanding of the rest of the manuscript. From the structure of the manuscript, I can understand that the authors tried to incorporate some more explanations into the results section to make them more readable and integrative, which normally should be done by discussing the data instead of providing background that should be stated in the introduction. While I am fine with the form of the manuscript, the authors should nevertheless explain a bit more the aims of this study based on specific literature, which is currently missing in the introduction, and instead stated in the results. By doing so this should bring the authors to better define the goals and objectives of the study, as well as the specific hypotheses tested in this study.

- There is clearly a lack of specific literature around the topic of the regulatory mechanisms of torpor and hibernation by (poly)unsaturated fatty acids (PUFA), notably n-6 PUFA, and the generation of oxidative stress. I strongly advice the authors to deepen the literature available on that specific topic, and especially on the fact that other hypotheses or explanations than the homeoviscous hypothesis might support the data observed concerning the effects of lipids on torpor and hibernation. There is a bunch of literature (see within my specific comments below) that the authors should read and probably cite as an alternative explanation.

Specific comments:

Introduction

- Page 4, line 38: maybe state that Syrian hamster is food-storing hibernator, i.e. feed in interbout arousals during hibernation

- Page 4, line 39: hibernation as torpor is a state of decreased metabolic rate (i.e. hypometabolism) associated with a state of reduced body temperature (i.e. hypothermia). This has its importance for the present study, as the reported observations are resulting from both a response to cold but also ischemia, a process which occur predominantly during torpor. This has to be stated.

- Page 4, line 43: through both non-shivering and shivering

- Information about differences between tissues (liver) is missing.

- The authors should also consider the resistance to ischemia during torpor which can also cause serious damages along with cold itself. Torpor is much more than a simply state of hypothermia of reduced temperatures!

Results

- Some acronyms are used without being previously defined, e.g. PI, LDH etc.

- Page 5, line 68: to which rate was the rewarming at 37°C? This matters to compare with the rate of natural rewarming of a hibernator from deep (~4°C) torpor.

- Page 6, lines 69-70: how does it compare to mouse hepatocytes?

- Page 6, line 83: introduce and tell shortly what is a STD diet?

- Page 7, line 85: introduce and tell shortly what is a STC diet?

- Page 7, lines 87-88: did you give STC diets to the mice?

- Page 7, line 93: define ROS

- Page 8, line 103: or can be and/or?
 - Page 9 line 123: in PE(18:0_20:4) 20:4 belongs to n-6 PUFA
 - Page 10, lines 142-146: could be stated already in the introduction
 - Page 11, line 158: in this study, HUFAs mostly contain n-6 PUFA (20:4), or?
- Discussion
- Page 15, line 204: mammalian deep hibernators
 - Page 15, line 207: that is change (increase) in metabolic rate which produces ROS, not body temperature!
 - Page 15, lines 208-209: as alternative of the homeoviscuous adaptation, there are other specific hypotheses in hibernation based on differential roles of n-6 and n-3 PUFAs (see Ruf and Arnold 2008 - doi:10.1152/ajpregu.00688.2007, Arnold et al. 2015 - doi:10.1152/physiol.00047.2014, Giroud et al. 2013 - doi:10.1371/journal.pone.0063111, Giroud et al. 2018 - doi: 10.3389/fphys.2018.01235, Logan et al. 2020 - doi: 10.3389/fmolb.2020.00097).
 - Page 15, line 211: in this study, HUFAs are mostly composed of n-6 PUFA?
 - General: how do the results relate to what hibernators, i.e. Syrian hamsters, experience in vivo during torpor, and along torpor-arousal cycle? Some more integrative approach including whole-body physiology would have been valuable to compare with the cell culture measurements.
- Methods
- Page 19. It is absolutely not clear how the animals were sacrificed and samples. Were the hamsters sampled during torpor, during interbout arousals (IBE)? What was the body temperature? Do the authors know how long the animals were in IBE?
 - Page 19, line 272: what are the compositions of the diets?
 - Page 19. The authors must consider and state that Syrian hamsters are food-storing hibernators, therefore expecting an impact of the diet during hibernation (in contrast to fat-storing hibernators than do not feed during their winter hibernation).
 - Page 20, line 281: actually C57BL/6 (black-6) mice are daily heterotherms and can enter daily torpor (especially when under cold exposure and/or fasted)! How do the authors know that mice were not in daily torpor, i.e. hypothermic and hypometabolic?
 - Page 20, line 283: the thermoneutral zone for mice is actually rather 27-29(-30)°C, hence being cold-challenged at 22-25°C!

Reviewer #2:

Remarks to the Author:

Anegawa et al. proposed molecular mechanism of cold resistance in cultured hepatocytes from Syrian hamster, a mammalian hibernator. The study presents several potentially novel and intriguing ideas, such as diet-dependent adaptation to cold.

In general, I believe that this body of work will appeal to the Communications Biology's broad readership, including those interested in cold adaptation, neurophysiology of hibernation, cell metabolism and mitochondria function.

There are several points which have to be addressed:

1. Page 4 line 45: "...then become hypothermic again by shutting off thermogenesis and becoming immobile." I believe this statement is not accurate, since hibernators maintain thermogenic ability. I would rephrase it: "by significant reduction in thermogenesis...".
2. Figure 2. It is not clear why authors used this statistical analysis. I recommend using One-way ANOVA multiple comparisons.
3. Methods section: please provide age and sex for mice. It is also not clear if hamsters had similar body weight across experimental groups, since this may affect cold resistance at the organismal and cellular levels.
4. Discussion: The authors did not put their finding in the context of other work. I suggest expanding discussion section and talk about synergistic (or not) mechanisms which were described by: 1. Jingxing Ou et al, Cell 2018, PMID: PMC5935596. 2. Neel S Singhal et al, eLife 2020, PMID: PMC7671683. 3. Koen D W Hendriks et al., Int J Mol Sci 2020 Mar 9;21(5):1864. doi:

10.3390/ijms21051864 and others.

Reviewer #3:

None

Responses to reviewers (Decision on manuscript COMMSBIO-20-3386-T)

We thank the referees and editor for the effort and time required to evaluate our manuscript and their constructive comments. We have responded to all of the comments and have marked modified sentences in the revised manuscript in red. Please see our point-by-point responses below:

Reviewer #1 (lipidomics, hibernation physiology):

In this study, the authors investigated the mechanisms responsible for cold resistance in Syrian hamster's hepatocytes, which exhibited remarkable resistance to prolonged cold culture. In contrast, hepatocytes of non-hibernating mice underwent cold-induced cell death showing hallmarks of ferroptosis, such as lipid peroxidation. This study further revealed a diet-dependent resistance of Syrian hamster's hepatocytes to cold- and drug-induced ferroptosis linked to the specific retention of alpha-tocopherol in the liver. Below the authors can find some major and other minor comments to help them improve their manuscript.

Major comment:

One major comment resides in the status of the introduction, which is actually missing important and key information necessary for the understanding of the rest of the manuscript. From the structure of the manuscript, I can understand that the authors tried to incorporate some more explanations into the results section to make them more readable and integrative, which normally should be done by discussing the data instead of providing background that should be stated in the introduction. While I am fine with the form of the manuscript, the authors should nevertheless explain a bit more the aims of this study based on specific literature, which is currently missing in the introduction, and instead stated in the results. By doing so this should bring the authors to better define the goals and objectives of the study, as well as the specific hypotheses tested in this study.

Thank you for your comment. We have revised the Introduction and Discussion sections to more clearly explain the background information and the aim of this study.

There is clearly a lack of specific literature around the topic of the regulatory mechanisms of torpor and hibernation by (poly)unsaturated fatty acids (PUFA), notably n-6 PUFA, and the generation of oxidative stress. I strongly advice the authors to deepen the literature available on that specific topic, and especially on the fact that other hypotheses or explanations than the homeoviscous hypothesis

might support the data observed concerning the effects of lipids on torpor and hibernation. There is a bunch of literature (see within my specific comments below) that the authors should read and probably cite as an alternative explanation.

We acknowledge that there are many studies based on the hypothesis that n-6 PUFA and/or n-6/n-3 ratio may affect the expression and phenotypes of torpor in several hibernators, including alpine marmots, golden mantled ground squirrels, 13 lined ground squirrels, and so on. However, our manuscript did not focus directly on this hypothesis. In addition, recent studies on arctic ground squirrels and Syrian hamsters did not support this hypothesis, making a controversial situation in which the hypothesis itself is challenged (Rice, 2021; Trefna, 2017). We expect that higher resolution analysis of lipids in tissues and more precise examination of causal relationships by manipulating metabolic enzymes for those lipid species would be absolutely necessary to test the hypothesis and eventually identify critical lipid species involved in torpor regulation. From the above reasoning, we do not find plausible reasons and contexts for adding a discussion on the role of n-6/n-3 PUFA in the regulation of torpor and oxidative stress in our revised manuscript. However, as we agree with your opinion that the role of specific PUFAs in hibernation is an important issue, we have added a brief explanation of the hypothesis in the introduction as follows:

Line. 64-67 in Introduction

“In addition, seasonal remodeling of lipid composition in tissues occurs in a diet-independent manner, while there are hypotheses that dietary fatty acids, in particular, the ratio of dietary omega-3/omega-6 fatty acids, may affect the expression and quality of torpor 8,9,10 2015.”

Specific comments:

Page 4, line 38: maybe state that Syrian hamster is food-storing hibernator, i.e. feed in interbout arousals during hibernation

Page 19. The authors must consider and state that Syrian hamsters are food-storing hibernators, therefore expecting an impact of the diet during hibernation (in contrast to fat-storing hibernators than do not feed during their winter hibernation).

Thank for this comment. We have added this explanation and discussed it in Introduction and discussion as follows:

Line.95-97 in Introduction

“Syrian hamsters are food-storing hibernators that store food in their nest during the pre-hibernation period and eat them in euthermic periodic arousal during the hibernation period, in contrast to fat-storing hibernators that do not eat food and rely on stored body fat during hibernation⁷.”

Line. 275-281 in Discussion

“Since α T and other vitamin E analogues act as chain-breaking antioxidants to prevent lipid peroxidation and are easily obtained from available foods in wild environments⁴⁸, maintaining higher level of α T in the liver and the plasma seems to be an ideal solution to these challenges of hibernation, particularly for food-storing hibernators that can ingest dietary α T during periodic arousal. Further studies will be needed to examine whether fat-storing hibernators such as ground squirrels also have superior hepatic capacity for α T storage.”

Page 4, line 39: hibernation as torpor is a state of decreased metabolic rate (i.e. hypometabolism) associated with a state of reduced body temperature (i.e. hypothermia). This has its importance for the present study, as the reported observations are resulting from both a response to cold but also ischemia, a process which occur predominantly during torpor. This has to be stated.

Page 4, line 43: through both non-shivering and shivering

The authors should also consider the resistance to ischemia during torpor which can also cause serious damages along with cold itself. Torpor is much more than a simply state of hypothermia of reduced temperatures!

Information about differences between tissues (liver) is missing.

Page 10, lines 142-146: could be stated already in the introduction

We have revised the introduction to add further explanation according to the above comments.

Some acronyms are used without being previously defined, e.g. PI, LDH etc.

We have added definition for acronyms. Thank you.

Page 5, line 68: to which rate was the rewarming at 37°C? This matters to compare with the rate of natural rewarming of a hibernator from deep (-4°C) torpor.

Thank you for your comment. Unfortunately, we did not measure the precise rate using a data logger or temperature probe. In the revised manuscript, we have conducted another experiment in which the cells were gradually and slowly cooled and rewarmed at a rate comparable to the *in vivo* situation (revised Fig 1D-G). This experiment clearly demonstrated that the speed of cooling and rewarming did not affect the survival rate of hamster hepatocytes. The revised sentences in the manuscript are as follows:

Line.122-126 in Results

“To examine whether the speed of cooling and rewarming may affect the survival rate of hepatocytes, the effect of the slow protocol on survival was also tested (Fig. 1F). No significant differences in the survival rate were found between the rapid and slow protocols (Fig. 1G), indicating that hamster hepatocytes *in vitro* have resistance to cold-rewarming stress.”

Page 6, lines 69-70. how does it compare to mouse hepatocytes?

It cannot be compared with mice, because mouse hepatocytes died only after 2 days of cold culture, making it impossible to conduct cold-rewarming experiments.

Page 6, line 83. introduce and tell shortly what is a STD diet?

Page 7, line 85. introduce and tell shortly what is a STC diet?

We have added the explanatory sentences as follows;

Line.141-143 in Results.

“The STD and STC diets are usually used to maintain Syrian hamsters and mice in our laboratory, respectively, and differ in their ingredients (see Methods).”

In addition, we have also supplied information of the ingredients in the two diets as Supplementary Table 2 (see later comment).

Page 7, lines 87-88. did you give STC diets to the mice?

We administered STD diets to the mice during the experiment. We have added a further explanation in red to the original sentences as follows:

Line. 143-146 in Results

“It should be noted, however, that the both mice and STD hamsters used in this study were fed an STD diet. Nevertheless mouse hepatocytes did not exhibit cold resistance (Fig. 1A, 1B, Fig. 2D). Thus, cold resistance in hepatocytes is exerted in an STD diet-dependent manner in hamsters, but not in mice”

Page 7, line 93. define ROS

Page 8, line 103. or can be and/or?

We have modified these points, thank you.

Page 9 line 123. in PE(18.0_20.4) 20.4 belongs to n-6 PUFA

Page 11, line 158. in this study, HUFAs mostly contain n-6 PUFA (20.4), or?

Page 15, line 211. in this study, HUFAs are mostly composed of n-6 PUFA?

The mass spectrum data obtained by LC-MS/MS analysis in negative mode showed characteristic peaks (m/z 303.23 and 283.26, respectively) for fatty acid 20:4 and fatty acid 18:0 (18:0_20:4) from PE (Fig. R1). Based on the fatty acid composition of mouse ^{1,2} and hamster ³ livers reported in the past, we can assume that 20:4 is n-6 arachidonic acid.

As for the HUFA of PC shown in Fig. 3, since LC-MS/MS was performed using cation analysis, fragment data of MS/MS spectra showing fatty acid composition were scarce. Therefore, only compositional formula identification (number of carbon chains and double bonds) by accurate mass using Orbitrap was performed, and the fatty acids in the sn-1 and 2-position fatty acids could not be qualitatively determined here.

However, according to the literature ^{1,2}, which examined the composition of fatty acid species in mouse liver, arachidonic acid (20:4 n-6) and DHA (22:6 n-3) are considered to be abundant as HUFAs. In this respect, in hamsters, little is currently known about hepatic fatty acid composition, and careful qualitative analysis is needed to determine the proportion of n-6 or n-3 fatty acids in each phospholipid. It should be noted that resolving these points is beyond the scope of this study.

Fig. R1 Mass spectrum data of PE(18.0_20.4) identified in lipidomic analysis

Page 15, line 204: mammalian deep hibernators

Page 15, line 207: that is change (increase) in metabolic rate which produces ROS, not body temperature!

We have modified these points, thank you.

Page 15, lines 208-209: as alternative of the homeoviscous adaptation, there are other specific hypotheses in hibernation based on differential roles of n-6 and n-3 PUFAs (see Ruf and Arnold 2008 - doi:10.1152/ajpregu.00688.2007, Arnold et al. 2015 - doi:10.1152/physiol.00047.2014, Giroud et al. 2013 - doi:10.1371/journal.pone.0063111, Giroud et al. 2018 - doi: 10.3389/fphys.2018.01235, Logan et al. 2020 - doi: 10.3389/fmolb.2020.00097).

Thank you for providing this information. However, in addition to the homeoviscous adaptation hypothesis, we have already provided an alternative explanation that differential PUFA profiles may contribute to differential sensitivity to cold-induced ferroptosis. Moreover, as mentioned before, the hypothesis that n-6 or n-6/n-3 PUFA may play a role in torpor is neither within the scope of nor directly assessed in this study. Thus, we have not been able to find a plausible reason to adapt it as an alternative

explanation of differential cold vulnerability between non-hibernators and hibernators, as well as between STC hamsters and STD hamsters.

General: how do the results relate to what hibernators, i.e. Syrian hamsters, experience in vivo during torpor, and along torpor-arousal cycle? Some more integrative approach including whole-body physiology would have been valuable to compare with the cell culture measurements.

Surprisingly, the STC hamsters hibernated, as well as the STD hamsters (Fig. R2). Interestingly, we found that plasma α T concentration in euthermic STC hamsters was higher at periodic arousal during the hibernation period than in the non-hibernation period, implying the importance of α T for cold resistance *in vivo* as follows:

Line248-251, in Results

“Interestingly, the plasma α T concentration of euthermic STC hamsters at periodic arousal during the HIB period was about five fold higher ($8.85\pm 3.23 \mu\text{M}$) than those in the non-HIB period ($1.57\pm 0.24 \mu\text{M}$) (Fig. 4F), implying the existence of systems to compensate for the low amount of α T taken from the diet during the hibernation period in Syrian hamsters.”

Thus, we have added the above data as whole body physiology data during hibernation in the revised manuscript. Unfortunately, comparison of α T concentrations between PA and DT could not be performed in this study because of the limited number of hibernating animals. Tb changes during hibernation were also preliminarily examined in a few animals (Fig. R2), although we did not analyze it quantitatively. To evaluate the role of effective storage of α T in hibernation physiology, a more refined strategy using a large number of animals, ingredient-precisely controlled diets, and time-course sampling during hibernation are necessary. It will take years and be reported elsewhere. Thus, we have added the following discussion:

Line. 310-335 in Discussion

“One limitation of this study was that the importance of α T in hibernation was not directly assessed *in vivo*. Although this study demonstrated that hepatocytes from hamsters fed an STC diet were

vulnerable to cold-induced ferroptosis under cultured conditions, we observed that the STC hamsters hibernated and survived (our unpublished observation). The discrepancy between the in vitro cold vulnerability and the in vivo cold resistance may be explained by the existence of mechanisms that compensate for the hepatic α T shortage during hibernation. One possible compensatory mechanism is that antioxidants, including vitamin C and vitamin E itself, are supplied systemically from other organs via the blood stream. In fact, we found that the STC hamsters in the hibernation period exhibited much higher plasma α T concentrations than those in summer-like conditions. In addition, it has been reported that the plasma concentration of vitamin C, which can regenerate α T radicals, changes markedly during cycles of PA and DT bouts in arctic ground squirrels and Syrian hamsters ⁵⁴⁻⁵⁶. In rodents, vitamin C is synthesized in the liver and other organs and may be supplied to the liver during hibernation. As such, a continuous supply of vitamin C from the blood may regenerate α T, thereby preventing its depletion and lipid peroxidation in cells and tissues during hibernation. Thus, retention and regeneration of α T may prevent cell death in a concerted manner in vivo during hibernation. This idea is also consistent with a previous report that plasma α T concentration increased during hibernation in Syrian hamsters ⁵⁶. Although it was implicated that high amount of dietary α T may inhibit torpor in golden mantled ground squirrels ⁵⁷, more intervention-driven studies using a large number of animals with time-course sampling during hibernation, ingredient-precisely controlled diets, and genetic manipulation will be necessary to evaluate the role of effective storage of α T in hibernation physiology ³. Another limitation of this study is that it is unclear at present whether the action of α T is parallel to or involved in in vitro cell-autonomous cold resistant mechanisms that have been proposed in other cell types and cancer cell lines derived from hibernators ^{19,22,23,25,26}. Since the in vivo significance of these mechanisms is also lacking, future studies on the mechanistic dissection of their functional contribution to and crosstalk in hibernation physiology are needed.”

[Redacted]

Methods

Page 19. It is absolutely not clear how the animals were sacrificed and samples. Were the hamsters sampled during torpor, during interbout arousals (IBE)? What was the body temperature? Do the authors know how long the animals were in IBE?

A method for sacrificing and sampling was described in the isolation of the hepatocyte section. For IBE animals, we sampled hibernating hamsters at the spontaneous interbout arousal (in our article, it is designated as periodic arousal) phase. Although we did not implant a Tb logger in these animals, it was evident that they were euthermic in that they actively moved immediately before introducing anesthesia. We have added the following explanation:

Line. 357-370

“The “saw-dust method” was also used for confirming that animals successfully hibernated; wood chips placed on the back of hibernating individuals remained in place until the animals experienced a periodic arousal.⁴ Under this condition, most animals start hibernation 2-4 months after cold exposure ⁵. For sampling animals in the PA phase during the HIB period, we observed animal status every morning by visual inspection with the sawdust method mentioned above to check whether the hibernating animals were in the PA or DT phase. Euthermic animals in spontaneous PA phases, judged by locomotion in cages and the reaction to handling, were anesthetized and used for hepatocyte culture around 12:00:16:00 (ZT2-6)(See “isolation of hepatocytes” for details). In this study, Tb and duration during PA were not determined, as a Tb logger (iButton) was not implanted into the animals used for hepatocyte culture because of the concern that surgical operation of Tb loggers into the body cavity might affect liver physiology and hepatocyte culture. Animals were sacrificed under anesthesia with 4.5% isoflurane for hepatocytes isolation by reperfusion or for blood collection by decapitation at 1318 weeks of age, except for Figs. 1D-E and 4F in which the animals were at 31-36 weeks of age.”

”

Page 19, line 272: what are the compositions of the diets?

The STD and STC diets differed in their composition, according to the manufacturer’s information (Supplementary Table. 2). The lipid profile was measured using GC analysis, as shown in the table. Although many ingredients differ in their amount between the two diets, we succeeded in identifying

α T as a factor necessary and sufficient for exerting cold- and cold-rewarming resistance in hamster hepatocytes.

A

	STC	STD
Water	9.2%	9.0%
Crude protein	18.8%	24.2%
Crude fat	3.9%	4.5%
Coarse fiber	6.6%	4.0%
Ash content	6.9%	6.5%
Nitrogen-free extract	54.7%	51.9%
Calorie	3291 kcal/kg	3449 kcal/kg
Vitamin A	16150 IU	15653 IU
Vitamin D3	3169 IU	2293 IU
Vitamin E	45.2 mg	208.3 mg
Vitamin K3	15.4 mg	27.4 mg
Choline	1820.0 mg	2948.0 mg
Folic acid	2.6 mg	5.6 mg
Niacin	101.7 mg	67.8 mg
Pantothenic acid	25.1 mg	41.1 mg
Biotin	0.3 mg	0.8 mg
Vitamin B1	10.9 mg	41.8 mg
Vitamin B2	8.9 mg	19.8 mg
Vitamin B6	18.9 mg	21.3 mg
Vitamin B12	0.026 mg	0.030 mg
Vitamin C	49.1 mg	1562.6 mg
Inositol	9.7 mg	741.2 mg
Carotene	1.2 mg	1.8 mg
Arginine	1.09%	1.27%
Histidine	0.46%	0.54%
Isoleucine	0.67%	0.89%
Leucine	1.36%	1.81%
Lysine	0.89%	1.09%
Methionine	0.26%	0.35%
Phenylalanine	0.84%	1.17%
Threonine	0.64%	0.80%
Tryptophan	0.21%	0.23%
Valine	0.79%	1.00%
Cysteine	0.28%	0.35%
Ca	0.98%	1.17%
Cl	0.41%	0.26%
Mg	0.26%	0.17%
P	0.80%	0.69%
K	0.97%	0.82%
Na	0.29%	0.22%
I	0.77 mg/kg	1.12 mg/kg
Fe	239.8 mg/kg	238.7 mg/kg
Co	0.08 mg/kg	0.12 mg/kg
Mn	101.2 mg/kg	68.1 mg/kg
Se	0.15 mg/kg	0.18 mg/kg
Zn	88.7 mg/kg	40.6 mg/kg
Cu	14.2 mg/kg	12.9 mg/kg

B

	STC	STD
Total FA	3.95%	4.84%
SFA	0.74%	0.94%
UFA/SFA ratio	4.33%	4.14%
MUFA/SFA ratio	1.21%	1.12%
MUFA	0.90%	1.06%
PUFA	2.31%	2.84%
HUFA	0.23%	0.35%
PUFA/MUFA ratio	2.56%	2.67%
n-3 PUFA	0.23%	0.35%
n-6 PUFA	2.08%	2.49%
n-6/n-3 ratio	9.04%	7.11%
C14:0	0.01%	0.02%
C14:1	N.D.	N.D.
C15:0	N.D.	0.01%
C16:0	0.65%	N.D.
C16:1	0.03%	0.02%
C17:0	N.D.	N.D.
C17:1	N.D.	N.D.
C18:0	0.07%	0.13%
C18:1	0.82%	0.98%
C18:2 n-6	2.08%	2.49%
C18:3 n-3	0.16%	0.26%
C20:0	0.01%	0.01%
C20:1	0.04%	0.05%
C20:5 n-3	0.03%	0.04%
C22:0	N.D.	0.01%
C22:1	0.01%	0.01%
C22:5	N.D.	N.D.
C22:6 n-3	0.04%	0.05%
C24:0	N.D.	N.D.
C24:1	N.D.	N.D.

Supplementary Table 2. Composition of diets used in this study.

Composition of ingredient according to information provided by the manufacture (A). Fatty acids composition analyzed by gas chromatography (B). SFA: saturated fatty acid. UFA: unsaturated fatty acid. MUFA: monounsaturated fatty acid. N.D. ; Not detected.

Page 20, line 281: actually C57BL/6 (black-6) mice are daily heterotherms and can enter daily torpor(especially when under cold exposure and/or fasted)! How do the authors know that mice were not in daily torpor, i.e. hypothermic and hypometabolic?

Page 20, line 283: the thermoneutral zone for mice is actually rather 27-29(-30)°C, hence being cold-challenged at 22-25°C!

We agree that room temperature of 22-25°C is mildly cold for mice. However, as they can access food and water ad libitum, we have never observed that the wild-type mice entered daily torpor under this condition in our laboratory. We can say this with confidence, although we did not monitor mouse Tb in this study, because we have also been studying mouse daily torpor and observed how they look in hypometabolic and hypothermic daily torpor states. Importantly, mouse hepatocytes were not cold-resistant even when the animals were exposed to mild cold.

References in the rebuttal letter

- 1 Puri, P. et al. A lipidomic analysis of nonalcoholic fatty liver disease. *Hepatology* 46, 1081-1090, doi:10.1002/hep.21763 (2007).
- 2 Wang, X., Cao, Y., Fu, Y., Guo, G. & Zhang, X. Liver fatty acid composition in mice with or without nonalcoholic fatty liver disease. *Lipids Health Dis* 10, 234, doi:10.1186/1476-511X-10-234 (2011).
- 3 Miranda, J. et al. Hepatomegaly induced by trans-10,cis-12 conjugated linoleic acid in adult hamsters fed an atherogenic diet is not associated with steatosis. *J Am Coll Nutr* 28, 43-49, doi:10.1080/07315724.2009.10719760 (2009).

Reviewer #2 (mammalian thermoregulation):

Anegawa et al. proposed molecular mechanism of cold resistance in cultured hepatocytes from Syrian hamster, a mammalian hibernator. The study presents several potentially novel and intriguing ideas, such as diet-dependent adaptation to cold.

In general, I believe that this body of work will appeal to the Communications Biology's broad readership, including those interested in cold adaptation, neurophysiology of hibernation, cell metabolism and mitochondria function.

There are several points which have to be addressed:

Page 4 line 45: "...then become hypothermic again by shutting off thermogenesis and becoming immobile." I believe this statement is not accurate, since hibernators maintain thermogenic ability. I would rephrase it: "by significant reduction in thermogenesis..."

Thank you for this comment. We have modified this as "by **significant reduction in metabolic rate and** thermogenesis."

Figure 2. It is not clear why authors used this statistical analysis. I recommend using One-way ANOVA multiple comparisons.

We have replaced the analysis by one-way ANOVA and the Tukey's multiple comparison test, in which statistical significance was also obtained between the control and drug-treated groups.

Methods section: please provide age and sex for mice.

We have added the information in Methods as follows;

Line. 371-376 in Methods

“Male C57BL/6 mice were purchased from SLC, Inc, Japan. Animals were reared under summer-like conditions (light condition = 14L:10D cycle, lights on 06.00-20.00, ambient temperature = 22°C - 25°C) and had ad libitum access to STD diet (MR standard diet, Nihon Nosan, Japan) and water in this experiment. Because we did not have information on the diet the mice had been fed in the breeding company, we purchased the mice at 8 weeks of age and fed the STD diet for over 2 months and used for hepatocyte culture experiments at 16-21 weeks of age.”

It is also not clear if hamsters had similar body weight across experimental groups, since this may affect cold resistance at the organismal and cellular levels.

We confirmed that body mass (weight) did not affect the difference of cold susceptibility between STD and STC. We have added a sentence and Supplementary Figure 1 as follows;

Line. 131-132 in Results,

“The cold resistance of the hamsters was not affected by body mass of the animals used for hepatocytes isolation (Supplementary Fig. 1).”

Supplementary Fig. 1 Relationship between the amount of cell death (%LDH) in hepatocytes exposed to cold culture for 48 h and body mass of the hamster used for hepatocytes isolation. Body mass did not contribute to differential susceptibility between STC and STD.

Discussion: The authors did not put their finding in the context of other work. I suggest expanding discussion section and talk about synergistic (or not) mechanisms which were described by: 1. Jingxing Ou et al, Cell 2018, PMID: PMC5935596. 2. Neel S Singhal et al, eLife 2020, PMID: PMC7671683. 3. Koen D W Hendriks et al., Int J Mol Sci 2020 Mar 9;21(5):1864. doi: 10.3390/ijms21051864 and others.

Thank you for your comment. We have added more information and discussion as for previous works. As for Singhal et al, the *ATP5G1* variants identified in the study was not found in hamsters, excluding it as a mechanism for cold resistance in hamsters. At present, we do not know whether other mechanisms operate in hamsters synergistically or not, but we have added that investigation of these points will be carried out in future:

Line73-78 in Introduction

“Several mechanisms underlying cold resistance have been proposed, including prevention of ROS generation in neurons differentiated from induced pluripotent stem cells of 13-lined ground squirrels ²⁴, H2S generation and sustained mitochondrial membrane potential in Syrian hamsters ^{19,22,25}, and Atp5G1 gene variants specific to arctic ground squirrels ²⁶. However, the in vivo significance of these mechanisms is still lacking, and the molecules responsible for cold resistance remain to be elucidated.”

Line.330-335 in Discussion

“Another limitation of this study is that it is unclear at present whether the action of αT is parallel to or involved in in vitro cell-autonomous cold resistant mechanisms that have been proposed in other cell types and cancer cell lines derived from hibernators ^{19,22,23,25,26}. Since the in vivo significance of these mechanisms is also lacking, future studies on the mechanistic dissection of their functional contribution to and crosstalk in hibernation physiology are needed”

REVIEWERS' COMMENTS:

Reviewer #1 (Remarks to the Author):

Dear authors,
many thanks for the thorough revisions yo have made on the manuscript, which reached my expectations.
Congratulations to this nice study!
All the best,
Sylvain Giroud

Reviewer #2 (Remarks to the Author):

The authors adequately addressed all my comments.